# Does Vitamin K2 Influence the Interplay between Diabetes Mellitus and Intervertebral Disc Degeneration in a Rat Model?

**DOI:** 10.3390/nu15132872

**Published:** 2023-06-25

**Authors:** Mohamed Mahmoud, Maria Kokozidou, Clemens Gögele, Christian Werner, Alexander Auffarth, Benjamin Kohl, Ingo Mrosewski, Gundula Gesine Schulze-Tanzil

**Affiliations:** 1Institute of Anatomy and Cell Biology, Paracelsus Medical University, Nuremberg and Salzburg, Prof. Ernst Nathan Str. 1, 90419 Nuremberg, Germany; mmagdibayoumi@gmail.com (M.M.); maria.kokozidou@pmu.ac.at (M.K.); clemens.goegele@pmu.ac.at (C.G.); c.werner@pmu.ac.at (C.W.); 2Department of Orthopedics and Traumatology, Paracelsus Medical University, Müllner-Hauptstraße 48, 5020 Salzburg, Austria; a.auffarth@salk.at; 3Department of Traumatology and Reconstructive Surgery, Charité–Universitätsmedizin Berlin, Corporate Member of Freie Universität Berlin, Humboldt-Universität zu Berlin, Campus Benjamin Franklin, Hindenburgdamm 30, 12203 Berlin, Germany; benjamin.kohl@charite.de; 4MVZ MDI Limbach Berlin, Aroser Alle 84, 13407 Berlin, Germany; ingo.mrosewski@mvz-labor-berlin.de

**Keywords:** diabetes mellitus type 2, intervertebral disc, intervertebral disc degeneration, fibrochondrocytes, Zucker Diabetic fatty (ZDF) rats, vit. K2-MK7, IL-10, IL-24, Hmox1, αSma

## Abstract

Intervertebral disc (IVD) degeneration is a common cause of low back pain in diabetes mellitus type 2 (T2DM) patients. Its pathogenesis and the vitamin (vit.) K2 influence on this disease remain unclear. Lumbar motion segments of male Zucker Diabetes Fatty (ZDF) rats (non-diabetic [control] and diabetic; fed without or with vit. K2) were used. Femur lengths and vertebral epiphyseal cross-section areas were measured. IVDs were histopathologically examined. Protein synthesis and gene expression of isolated IVD fibrochondrocytes were analyzed. T2DM rats showed histopathological IVD degeneration. Femur lengths and epiphyseal areas were smaller in T2DM rats regardless of vit. K2 feeding. Fibrochondrocytes synthesized interleukin (IL)-24 and IL-10 with no major differences between groups. Alpha smooth muscle actin (αSMA) was strongly expressed, especially in cells of vit. K2-treated animals. Gene expression of aggrecan was low, and that of *collagen type 2* was high in IVD cells of diabetic animals, whether treated with vit. K2 or not. *Suppressor of cytokine signaling* (*Socs*)*3* and *heme oxygenase* (*Hmox*)1 gene expression was highest in the cells of diabetic animals treated with vit. K2. Vit. K2 influenced the expression of some stress-associated markers in IVD cells of diabetic rats, but not that of IL-10 and IL-24.

## 1. Introduction

### 1.1. Intervertebral Discs

Intervertebral discs (IVDs) make up about 20–30% of the spine length [1]; function as shock distributers, load cushions, stress dispersers and weight carriers; and vary in size and constitution according to their anatomical locations, spinal level, the organism’s age and biomechanical demands/functions [1,2,3,4]. IVDs are avascular and receive nutrition from the adjacent blood vessels through diffusion, which is a contributing cause of degeneration [1]. An annulus fibrosus (AF), composed of water, collagen (types 1 and 2), proteoglycans (PG) and extracellular matrix (ECM) proteins with two main layers, encloses the central nucleus pulposus (NP) [1,5]. Fifteen to twenty-five distinct layers, called lamellae, in turn constitute the outer and inner zones of the AF [1,2]. The NP is a gelatinous mass, composed of 80–90% water and incompressible [1,3]. In addition to collagen type 2 fibrils, PGs hydrogel and elastin fibers, it contains few chondrocyte-like cells [1,6], which are metabolically active in the turnover process of the ECM proteins. During ageing, these cells become gradually inactive, less proliferative with impaired regenerative potential and necrotic, leading to continuous degradation of ECM contents. Additionally, the NP itself becomes more cartilaginous and less gelatinous [1,4]. The upper and lower vertebral endplates (VEPs) are osseo-hyaline-cartilaginous structures [1]. Considered the strongest components of the IVDs, they function as shields preventing the AF and NP from yielding under applied pressure and loads through “mechanical interlocking”. Additionally, the vascularized osseous parts of the VEPs have a nutritive role in the IVDs [1]. According to Zhang et al., the murine IVD exhibits, in the sagittal histological sections beginning from its center towards its upper/cranial or lower/caudal boundary: NP, AF, tidemark (TM), cartilage endplate (CEP) with 3–7 chondrocyte layers, growth plate (GP) and lower border of the overlying vertebral body (VB) cranially or upper border of the underlying VB caudally [7].

### 1.2. Low Back Pain, Intervertebral Disc Degeneration and Diabetes Mellitus Type 2

With ageing, IVDs degenerate, becoming more avascular, innervated, harder and less elastic, which contributes to low back pain (LBP) [1,4,8]. Genetic, metabolic, mechanical, environmental and ageing factors and lifestyle conditions were discussed as suspected underlying causes of IVD degeneration (IVDD) [1,2,5,6,8]. IVDD can be classified into five or eight grades in the classical and recently modified Pfirrmann’s grading systems according to MRI findings (MRI Classification System) [1,9].

IVDD is one of the primary causes of LBP [1,2], mostly appearing in the lumbar spine because of its high mobility and high load [1]. LBP is associated with chronic suffering and decrease of physical ability, leading to serious health and socioeconomical complications [2,5]. Globally, lumbar disc degeneration, low back pain (LBP) and T2DM are major chronic public health problems, mostly occurring concurrently [10,11].

Obesity and T2DM are considered to be the most relevant metabolic disorders leading to IVDD [2]. T2DM causes macrovascular, microvascular and connective tissue disturbances, leading to degenerative changes in the IVD and its close neighbors [2,12]. T2DM represents about 85–90% of diabetic cases and is caused mainly by insulin resistance at the receptor level or the progressive decrease of insulin secretion [2,13]. In 2017, 425 million cases of T2DM were reported worldwide [2]. Park et al. and Zheng et al. claimed that the pathomechanism of IVDD in diabetic cases is still unclear [11,14].

In a previous comprehensive review, we verified a link between T2DM and IVDD. However, many aspects (e.g., the pathogenesis) still need to be further explored [15].

Zheng et al. theorized that an amyloid peptide and interleukin (IL)-1β were the main causes of IVDD in T2DM [14]. In the long run, hyperinsulinemia, uncorrected hyperglycemia and the continuous production of advanced glycation end-products (AGEs) destroy IVD cells, leading to IVDD and/or IVD disc herniation [2,16]. The accumulation of AGEs, e.g., in NP proteins, including aggrecan and collagen, causes stiffening of the ECM through dysregulated collagen cross-linking, fibrosing, dehydration, degeneration and inhibition of repair and turnover [13]. Induction of reactive oxygen species (ROS) and apoptosis are typical features of hyperglycemia [17]. The complement system is also implicated in IVDD, with the cytoprotective complement regulatory proteins CD55 and CD59 being specific inhibitors of this cascade [18].

Mutations of the IL-10 gene were implicated in IVD degeneration [19,20,21]. Sudhir et al. linked IVDD in diabetic patients to the production and accumulation of ROS, causing oxidative stress, DNA damage, cellular apoptosis, altered gene expressions and production of inflammatory cytokines, growth factors and degradative enzymes [22]. Heme oxygenase-1 (HMOX1) is implicated in protection against tissue injury [23]. HMOX1 seems to attenuate IVD degeneration [24,25]. Its role in IVDs exposed to a diabetic environment remains unclear. It is upregulated by anti-inflammatory IL-10 and downregulated by pro-inflammatory cytokines [26]. Pro-inflammatory cytokines are also inhibited by suppressors of cytokine signaling (SOCSs) in a IVDD model providing a negative feedback loop [27]. We have demonstrated the chondroprotective properties of IL-10 [28,29]. However, the role of IL-24 has not been investigated so far. Additionally, it remains unclear whether IL-10 family cytokines and IVDD are influenced by high systemic serum vit. K2 levels.

### 1.3. Interrelation between Vitamin K, Diabetes Mellitus Type 2 and Intervertebral Disc Degeneration

Vitamin K (vit. K) is a highly complex compound and has many functions in the vital human body [30]. Vit. K has many biological roles, such as anti-inflammatory effects, regulation of calcium metabolism in tissues, protection from oxidative stress, regulation of cell growth and proliferation [30]. Vit. K regulates the mineralization of the ECM, protects against bone resorption and enhances bone formation through the inhibition of osteoclasts and stimulation of the osteoblasts, respectively [30]. Vit. K is considered a potent antioxidant agent protecting ECM and cell membranes from ROS and reactive nitrogen species (RNS) [31]. Vit. K is related to glucose metabolism and insulin sensitivity, but the effect of sufficient intake of vit. K on T2DM is still in need of further evaluation [32,33]. However, Ho et al. reported the obvious low risk of T2DM with sufficient vit. K intake, in addition to improvement of insulin sensitivity [34]. Additionally, Karamzad et al. linked vit. K administration to significantly lowered blood glucose and increased fasting serum insulin levels in diabetic animal models [35]. Further studies reported a lower incidence rate of T2DM with sufficient serum levels of vit. K [36,37,38]. The effect of vit. K on IVDD under the conditions of T2DM has not been elucidated so far.

Vit. K occurs as the fat-soluble vit. K1 and K2, while vit. K3 is synthetic and water-soluble [30]. Vit. K2, collectively termed “menaquinones”, is divided into short-chain menaquinones, e.g., vit. K2-MK4, and long-chain menaquinones, such as vit. K2-MK6 [39,40]. Vit. K2-MK7 is biologically more active than vit. K2-MK4 [39] and has higher absorbability as well as longer half-life than vit. K1. It exhibits about 70% of its activity in the extra-hepatic organs/tissues, such as femur, brain, testis, kidney and pancreas [41]. Vit. K2-MK7 is mainly produced by the bacterial fermentation of many food species, e.g., natto, soy, cabbage and cheese [39,40]. The usual daily physiological intake of Vit. K2-MK-7 is about 50–150 μg [41]. Vit. K2-MK7 is considered as a cofactor for activation and carboxylation of extrahepatic vit. K-dependent proteins (VKDPs) in the daily nutritional intake [42].

The aim of the present study was to analyze the effect of a vit. K2-MK7 diet on IVD histopathology and IVD cell phenotype in a T2DM rat model. Additionally, the expression profile of cartilage-specific components, IL-10 family cytokines and cell-stress-related markers in fibrochondrocytes derived from non-diabetic and diabetic rats were investigated.

## 2. Materials and Methods

### 2.1. Study Design

This project combines in situ and in vitro analyses of a diabetic rat model (ZDF rats).

#### Rat ZDF Diabetic Model

The diabetic rat model used in this study here is based on male Zucker Diabetes Fatty (ZDF) Leprfa/Crl rats (Fatty fa/+ and Fatty fa/fa, 4–7-month-aged, Charles River Laboratories International, Inc., Sulzfeld, Germany). Under a specific diabetogenic diet (Purina 5008, Firma sSniff, Spezialdiäten GmbH, Soest, Germany), only the homozygous male ZDF rats (fa/fa) developed T2DM after 12 weeks, whereas the heterozygous rats (fa/+) did not develop T2DM under similar conditions and served as controls. Vit. K2 (MK-7) (Kappa Bioscience AS, Oslo, Norway) was mixed in the diet at a level 100 mg/kg and fed to 95–110-day-old rats for 80 days until sacrifice. The onset of T2DM was detected by weekly blood sugar measurements using the AlphaTRAK veterinary measurement system (Zoetis, Tullytown, PA, USA). All animals were kept in standardized living conditions in a special incubator (55% humidity, 21 ± 2 °C; Memmert GmbH and co.KG, Schwabach, Germany) and fed according to their subgroups.

Animal experimentation protocols were approved by the local animal review board (RUF 55.2.2-2532-2-729-17). After sacrifice, the explanted whole lumbar vertebral column segments were subjected to histopathological analysis (staining with hematoxylin–eosin (HE), Alcian blue (AB) and Sirius red (SR) stain).

IVD-derived fibrochondrocytes were isolated according to [43,44] as described in detail in Section 2.3. Cells underwent immunohistological (confocal laser scanning microscope [CLSM], TCS SPEII; Leica, Wetzlar, Germany) and molecular (RNA isolation and real-time detection [RTD] PCR) investigations.

This study included 36 male animals (4–6 months old) divided into four subgroups (Table 1).

### 2.2. Preparation and Staining of Histological Samples

After the sacrifice of the animals in compliance with the animal protection FELASA guidelines, the whole motion segments (lumbar segments) of all animals were resected. For histology, IVDs were fixed in 4% paraformaldehyde (PFA/PBS, Santa Cruz Biotechnology Inc., Dallas, TX, USA) for 48 h. After that, bone decalcification was undertaken using EDTA solutions (PFA/PBS, Santa Cruz Biotechnology Inc., Dallas, TX, USA) for 4 weeks. Afterwards, tissue samples were cut in half along the median plane and embedded into paraffin (Path Center: Shandon Pathcenter, Thermo Scientific, Waltham, MA, USA) before dissection of 7 µm thick sections. Sections were fixed on the glass slides and kept in an oven (Memmert GmbH and Co.KG) at a temperature of 60 °C overnight. After being washed in xylol (Carl Roth GmbH and Co.KG, Karlsruhe, Germany) for deparaffinization and rehydration in descending sequence of ethanol (ETOH) concentrations (99.8%, 96%, 80%, 70%) (Carl Roth GmbH and Co.KG), the specimens were stained with hematoxylin–eosin (HE), Alcian blue (AB) and Sirius red (SR). After the staining, the sections were dehydrated in an ascending ETOH series (70%, 80%, 96% and 99.6%) and embedded in Entellan (Merck KGaA, Darmstadt, Germany). For examination, the 40× magnification of a light microscope (DM1000 LED with integrated camera, Leica Microsystems GmbH) or the polarized light modus of the CLSM were used.

#### 2.2.1. Hematoxylin–Eosin (HE) Stain

After incubation for 6 min in Harry’s hematoxylin (Carl Roth GmbH and Co.KG), the prepared IVD sections were rinsed in running tap water and counterstained with eosin (Carl Roth GmbH and Co.KG) for 4 min.

#### 2.2.2. Alcian Blue (AB) Stain

Alcian blue (AB) staining indicates the deposition of sulphated glycosaminoglycans in blue. For AB staining, the deparaffinized sections were incubated for 3 min in 1% acetic acid (Carl Roth GmbH and Co.KG) and then immersed for 30 min in 1% AB in acetic acid (Carl Roth GmbH and Co.KG). Subsequently, they were rinsed in 3% acetic acid. Cell nuclei were counterstained for 5 min with nuclear fast red aluminum sulphate solution (Carl Roth GmbH and Co.KG).

#### 2.2.3. Sirius Red (SR) Stain

Sirus red (SR) staining was performed to depict collagen distribution. Deparaffinized sections were washed in nondistilled (nondist.) water for 4 min. Cell nuclei were stained with Weigert’s hematoxylin (MORPHISTO GmbH, Frankfurt am Main, Germany) for 8 min. Afterwards, they were washed with dist. water for 5 s, followed by rinsing in nondist. water for 10 min and in dist. water for 1 min. Then, the sections were stained using SR (MORPHISTO GmbH) for 60 min, before being incubated twice in 30% acetic acid (Carl Roth GmbH and Co.KG) for 1 min each and then twice in 96% EtOH for 4 min.

#### 2.2.4. Measurement of Epiphyseal Cross-Section Areas

Alcian-blue-stained sections of all 4 animal groups (Table 1) were scanned using a scanning light microscope (Fritz, PreciPoint GmbH, Freising, Germany). Cross-section areas of both epiphyses of lumbar motion segments were measured using the PreciPoint analysis tool. For each animal, mean values of motion segments were calculated. 

#### 2.2.5. Histopathological Scoring System to Assess IVD Degeneration

Sections produced in the sagittal plane were assessed because degenerative changes of both AF and NP in IVD can be shown, according to Le Maitre et al. [45]. The most popular IVDD scoring system developed by Rutges et al. was selected because of its comprehensiveness, reliability and applicability [46]. The Rutges scoring system evaluates the degenerative changes in the following six aspects of IVD: CEP, AF, AF/NP border (AF/NP-B), NP cellularity (NPC), NP-ECM distribution (NPMD) and NP morphology (NPM) [5,47], as shown in Table 2. Each region was scored from 0 to 2 according to the degree of its degeneration: a score of 0 is healthy, 1 is mild to moderate and 2 is severely degenerated, and the total score for IVDD ranges from 0 to 12 according to the degree of health or disease; a total score of 0 is completely healthy and a total score of 12 is completely degenerated [5,47]. However, NP morphology includes two items (NP area and NP shape); hence, NP morphology was scored from 0 to 4 in our study and the total score ranged from 0 to 14. We did not find a suitable grading scale based on Rutges’ scoring system on medical websites and in electronic databases. Therefore, we generated a grading scale based on Rutges’ scoring system: 0 → IVD is fully healthy; 1–6 → mild degeneration; 7–9 → moderate degeneration; 10–14 → severe degeneration. Criteria of degeneration and scoring are depicted in Table 2. The scoring system of Rutges et al. is designed for human IVD [46]. We found that the cell distribution in the NP differs in human and rat IVD with naturally occurring cell groups in the center of the NP of rats. We also observed chondrocyte hypertrophy in the epiphyseal cartilage as a sign of degeneration, which was included in our adapted scoring system.

### 2.3. Rat IVD Cell Isolation and Cultivation

Representative IVDs were macroscopically visualized (Canon G9 X, Canon, Öta, Tokyo, Japan). IVD cells (fibrochondrocytes) were isolated from the ZDF rats of each group. After removing the surrounding ligaments and connective tissues, IVDs were sliced into 1 µm sized fragments and the IVD ECM was digested with 1 mg/mL collagenase (collagenase NB5 derived from *Clostridium histolyticum*, Nordmark, Uetersen, Germany), diluted in chondrocyte growth medium (96% [*v*/*v*] DMEM/Ham’s F-12 [1:1] with stable L-glutamin, 1% [*v*/*v*] amphotericin B, 1% [*v*/*v*] MEM amino acids, 1% [*v*/*v*] penicillin/streptomycin, 1% [*v*/*v*] ascorbic acid supplemented (all products from Carl Roth GmbH and Co.KG), containing 1% fetal bovine serum (FBS, Pan-Biotech GmbH, Aidenbach, Germany)) for 16 h in a TubeSpin bioreactor 15 mL tube (TPP, Trasadingen, Switzerland) at 37 °C under rotation (36 rpm). Subsequently, released cells were rinsed with growth medium with 10% FBS, counted and placed into a T75 flask (CellPlus, Sarstedt AG, Nümbrecht, Germany) with 10 mL of the above-mentioned growth medium. The culture medium was changed every 2–3 days for a total of 14 days. After cell confluence of about 80%, cells were detached with 0.05% trypsin and 0.02% ethylenediaminetetraacetic acid (EDTA) solution (Bio&SELL, Feucht, Germany). Cell viability and number of cells was calculated with a hemocytometer using the trypan blue exclusion assay and 10.000 cells/cm^2^ were seeded into T25 flasks (CellPlus, Sarstedt AG) and cultured for 48 h for RNA isolation.

### 2.4. Immunofluorescence Labeling for Protein Expression Analysis in IVD Fibrochondrocytes

To detect protein synthesis of cultured fibrochondrocytes from control and T2DM animals, the immunofluorescence labeling procedure was carried out, and then the immunoreactivity was assessed with CLSM. First, fibrochondrocytes (at 2nd–3rd passage) cultured on cover slides at 14,286 cells/cm^2^ for 48 h were fixed in 4% PFA for 15 min. Then, they were rinsed three times in TRIS-buffered saline (TBS: 0.05 M TRIS, 0.015 M NaCl, pH 7.6) and incubated with protease-free blocking solution (5% donkey serum diluted in TBS with 0.1% Triton × 100 for cell permeabilization) for 20 min at room temperature. Fibrochondrocytes were afterwards incubated with the following primary antibodies for one night and diluted in blocking solution at 4 °C in a humid incubator/box: [collagen type 1 (1:30, goat-anti-human), collagen type 2 (1:50, rabbit-anti-human), SRY-box transcription factor 9 (SOX9, 1:100, rabbit-anti-human), decorin (1:50, rabbit-anti-human), IL-10 1:30, (rabbit-anti-human), IL-24, aggrecan (1:30, mouse-anti-human) and α-smooth muscle actin (αSMA, 1:50, mouse-anti-human) (Table 3)].

Staining controls were similarly prepared, but the primary antibodies were omitted. Before incubation with the secondary antibodies, cells were rinsed three times with TBS. Then, they were incubated with the secondary antibodies coupled with (cyanine[cy]3-donkey-anti-mouse or -anti-goat in addition to donkey-anti-rabbit coupled with Alexa-Fluor488] diluted 1:200 in TBS at 4 °C for 1 h at room temperature. For visualization, cell nuclei were stained with 10 µg/mL 4′,6′- diamidino-2-phenylindol (DAPI, Roche, Mannheim, Germany). To depict the actin cytoskeleton, fibrochondrocytes were stained with phalloidin Alexa-Fluor488 (Table 3) and DAPI diluted in blocking buffer. Immunolabeled cells were then rinsed in TBS 3 times for 5 min before being covered with Fluoromount G (Biozol Diagnostica Vertrieb GmbH, Eching, Germany). CLSM was used to take photos of the immunolabeled cells. With a magnification of 630×, three separate photos were captured for each specimen, and then these photos of different microscopic fields were delivered to the software “Image J 152d” for their splitting into 3 channels (blue, red and green) and for the calculations of their fluorescence intensity. The used version was (1.53n/7 November 2021: ImageJ bundled with 64-bit Java 1.8.0_172 (70MB)).

### 2.5. RNA Isolation from Fibrochondrocytes

The IVD-derived fibrochondrocytes were sub-cultured to achieve sufficient cell numbers. Fibrochondrocytes, passages 1–3 of T2DM and control rats were cultured for 48 h at 10.000 cells/cm^2^ in T25 flasks and lysed in RLT buffer (Qiagen GmbH, Hilden, Germany) with 0.1% mercaptoethanol for about 15 min. Isolation and purification of RNA was achieved using the RNeasy Mini kit according to the manufacturer’s instructions (Qiagen GmbH) including an on-column DNA isolation via DNAse. Purity and quantity of the isolated RNA samples were evaluated at the 260/280 absorbance ratio using the Nanodrop ND-1000 spectrophotometer (Peqlab, Biotechnologie GmbH, Erlangen, Germany).

### 2.6. Quantitative Realtime PCR Analysis

Isolated total RNA (125 ng) was reversely transcribed into cDNA using the QuantiTect Reverse Transcription Kit (Qiagen GmbH) according to the manufacturers’ instructions. The TaqMan Gene Expression Assay (Life Technologies, Carlsbad, CA, USA) was used. As postulated in (Table 4), specific primer pairs were used for Hypoxanthine-Guanine-Phosphoribosyltransferase (*Hprt1*) as a reference gene, *aggrecan* (*Acan*), *Cd55*, *Cd59*, *collagen type 1 alpha 1 chain* (*Col1a1*) and *collagen types 2 alpha 1 chain* (*Col2a1*), *heme oxygenase 1* (*Hmox1*) and *suppressor of cytokine signaling 3* (*Socs3*). RTD-PCR was performed using the real-time PCR detector StepOnePlus (Applied Biosystems [ABI], Foster City, CA, USA) thermocycler with StepOnePlus software 2.3 (ABI). The mean normalized expressions (NME) of the genes of interest were assessed related to the reference gene and calculated for each sample, relying on the method introduced by Schefe et al. [48].

### 2.7. Statistics

The mean (MN), median (MD), standard deviation (SD), probability values (*p*-value), highest value (HV) and lowest value (LV)) were calculated to evaluate the histopathological, immunological and molecular results statistically. Statistics were performed with GraphPad Prism8 (GraphPad software Inc., San Diego, CA, USA). The analysis of the normal distribution of the results was achieved using the Shapiro–Wilk test. For comparison between groups, a two-tailed one-way ANOVA (Fisher) was used, followed by Tukey’s multiple comparison post hoc testing. The level of significance/confidence interval (CI) was set at *p* values of ≤0.05 (*), ≤0.01 (**), ≤0.005 (***) and *p* ≤ 0.001 (****). The test’s power was 0.8. The comparisons were undertaken between the control subgroup without vit. K2, control subgroup with vit. K2, diabetic subgroup without vit. K2 and the diabetic subgroup with vit. K2.

## 3. Results

### 3.1. Overall Performance and Behavior of the Rats

The diabetic animals showed typical features of T2DM, such as polydipsia, polyphagia, polyuria, hyperglycemia, obesity, untidy fur delayed wound healing, development of infections (mainly periodontal infections) and cataracts, as well as, a reduced overall activity compared to the control animals housed under similar conditions.

### 3.2. Incidental/Coincidental Macroscopic Findings

During the preparation of specimens, we noticed that the femur lengths of the diabetic rats were generally much shorter than those of the non-diabetic rats (Figure 1). Supplementation with vit. K2 had no significant effect on femur lengths in non-diabetic or in diabetic rats. Epiphyseal cross-section areas similarly were smaller in diabetic rats compared to non-diabetic animals, with vit. K2 having no influence.

### 3.3. IVDD Evaluation and Scoring

IVDs of heterozygous animals were characterized by regular morphology of the IVD (normal sizes of the NP with regular distribution of its lamellar ECM without debris and the NP being well integrated into the AF; normal concentric appearance of the lamellae of the AF as depicted by collagen stain with SR without gaps or disruption were visible, and no bony sclerosis and no obvious thinning of the epiphysis) (Figure 2).

In contrast to the heterozygous rats, diabetic animals showed disruption of the lamellae of AF, with the presence of clefts between them losing concentric appearance (Figure 2C,D). Clefts surrounding the NP were also detectable. In addition, more condensated NP ECM and cell groups could be detected in the T2DM rats compared to controls. Epiphyseal chondrocyte hypertrophy became evident in the IVDs of several, mainly diabetic animals. The mean score values for the IVDs of the diabetic animals, especially of those without vit. K2 treatment, were higher than of those IVDs derived from controls. The difference between the histological score values of controls and diabetic animals without or with vit. K2 treatment was significant (Figure 3).

### 3.4. Cell Isolation from the IVDs

Lumbar IVDs were dissected and explanted. The IVDs containing the NP (arrow, Figure 4A–D) used for cell isolation are shown in Figure 4. No macroscopical differences between the IVDs of the different groups were detected. The isolated and cultured IVD cells had a typical chondrocyte-like appearance with a cuboid shape and became more fibroblast-like during culturing (Figure 4E–H).

### 3.5. IVD-Derived Fibrochondrocyte-Related Protein Synthesis

Since the cells were permeabilized, the intracellular pool of ECM proteins was also immunolabeled and visualized (Figure 5). Collagen type 1 is a main component of IVDs. It was predominately intracellularly expressed in the perinuclear rough endoplasmic reticulum (rER) region of IVD cells, but not all cells were immunoreactive for collagen type 1 (Figure 5(B1,D1), blue arrows). Sometimes, procollagen accumulated in larger vesicles and its secretion could be visualized by immunolabeling. Some extracellular fibers of mature collagen type 1 were also detectable (red arrow, Figure 5(B1,B3)). The differences of collagen type 1 expression between groups were not statistically significant. All IVD cells were positive for the chondrocyte marker collagen type 2. Extracellular collagen type 2 fibrils and networks covering the cells could be seen in many samples (Figure 5(B2,B3,D2,D3), green arrow). Cells which were negative for collagen type 1 were often strongly positive for type 2 collagen. No statistically significant differences in collagen type 2 expression became evident when comparing the fibrochondrocytes of the different groups (Figure 5). With regard to collagen type 1 expression, there was a trend of more expression in cells of T2DM animals without vit. K2 supplementation.

The SOX9 immunoreactivity staining displayed a cytoplasmic and/or nuclear distribution. All cells were positive for SOX9. There was only a weak signal for cartilage PGs, but not all cells were immunoreactive for PGs. PGs had a cytoplasmic and/or extracellular distribution. Neither statistically significant differences nor an obvious trend of PG and SOX9 immunoreactivity in relation to T2DM with/without vit. K2 treatment could be detected (Figure 6).

The large PG aggrecan and the small PG decorin are typical components of the IVD. Aggrecan displayed a weak cytoplasmic and perinuclear signal. Most cells were at least weakly positive for it. There was a weak signal for decorin and not all cells were positive for it. A tendency of lower aggrecan expression in IVD cells from diabetic rats without vit. K2 supplementation or non-diabetic animals treated with vit. K2 became evident (Figure 7).

IVD cells were generally weakly immunoreactive for the anti-inflammatory cytokine IL-10. In some cultures, it was confined to the perinuclear region or reflected a faint cytoplasmic distribution. More intense and lesser immunoreactive cells could be found in the same culture. In some cells, the release of exosomes immunoreactive for IL-10 was detectable (Figure 8). There was a high variability in staining intensity of IL-10, with no statistically significant differences between the groups, but with a trend of higher expression in cells of diabetic donors.

IVD cells were weakly immunoreactive for the cytokine IL-24 (Figure 8(A2–D2)). The staining showed a cytoplasmic distribution. Sometimes, the staining was more intensive in the intra- and extracellular vesicles released by the cells (Figure 8(A2–D2,A4–D4), green arrows). There was no statistically significant difference in the staining intensity of IL-24 between cells from non-diabetic or diabetic donors irrespective of whether they were fed without vit. K2 or with vit. K2 supplementation. All cells showed at least a weak intracellular cytoplasmic immunoreactivity for αSMA, irrespective of their origin from non-diabetic or diabetic animals either fed without or with additional vit. K2 (Figure 8(A3–D4)). Several cells showed more intensively stained intracellular plaques strongly immunoreactive for αSMA (Figure 8G). Singular cells possessed highly αSMA immunoreactive stress fibers (Figure 8(A3,A4),G, thick white arrows). Interestingly, the mean fluorescence intensity for αSMA was higher in cells from animals fed with vit. K2 supplementation, irrespective of whether they were not diabetic or diabetic (Figure 8F).

### 3.6. Fibrochondrocytes Gene Expression

Gene expression of the typical IVD ECM components *collagens types 1*, *2* and *aggrecan* could be detected in the cultivated IVD cells. A lower level of *collagen type 1* and *aggrecan* gene expression was found in the cells of homozygous diabetic donors (Figure 9A,C), especially when treated with vit. K2 in the case of *aggrecan*, whereas *collagen type 2* was more highly expressed in the homozygous diabetic rat (Figure 9B). However, none of these differences were statistically significant.

Gene expression of cytoprotective components, such as complement regulatory proteins *Cd55* and *Cd59*, as well as *Hmox1*, was also shown. *Socs3* gene expression was detected. In homozygous diabetic rats, *Cd55* and *Socs3* were expressed lower than in heterozygous rats, but this was not the case with regard to *Cd59* and *Hmox1*.

In the IVD cells of homozygous rats, vit. K2 supplementation was associated with a higher gene expression level of cytoprotective factors (*Cd55*, *Cd59*, *Hmox1*, *Socs3*). However, in the heterozygous rats, there were lower mean values of *Cd59*, *Hmox1* and *Socs3* detectable in IVD cells of vit. K2-supplemented animals.

## 4. Discussion

Diabetic rats showed typical clinical features of T2DM. Clinical effects or side effects of vit. K2 on the rats could not be detected. Known side effects of vit. K2 include allergic reactions, jaundice, hepatotoxicity and hemolytic anemia in babies [49,50], overcoagulation, paradoxical hypoprothrombinemia, hemorrhages, anemia, minor gastrointestinal complaints [51], bronchospasm and cardiac arrest [52].

To understand the difference in the exterior of the homo- and heterozygous rats, the length of the femurs was compared after sacrifice. Measurements showed that femurs were significantly shorter in diabetic compared to non-diabetic rats with no difference between animals receiving or not receiving additional vit. K2 with their feed. Long bones, such as femurs and tibiae, increase in length through their proximal and distal epiphyseal growth plates and in width/thickness through their periosteal layers in a process called appositional growth. Chondrocytes and osteoblasts are responsible for a long bone’s lengthening and thickening, respectively [53,54,55]. The actual underlying cause of the femoral shortening is unclear. It could be that this finding arises from metabolic disturbances accelerating premature epiphyseal closure in diabetes or is a side effect of the gene defect causing the leptin receptor deficiency [56] leading to T2DM. In terms of the pathomechanism, the continuously elevated blood sugar levels, the accumulation of other diabetic metabolites and the endangering of vascular channels may harmfully affect the proximal and/or distal femoral epiphyseal growth plates, leading to chondrocyte apoptosis and epiphyseal failure [57]. We could not find any published reports which discuss if vit. K2 is harmful or useful for the epiphyseal growth plates. We discussed in the introduction that vit. K is considered to have an osteoblast-stimulatory effect. However, this osteoblastic effect is related to the ordinary growth (turnover) and remodeling of bones after epiphyseal closure and during the process of fracture healing, and not related to the continuous bone growth in children and teenagers until the fusion of the ossification centers between the ages of 18–25 years. Some papers have discussed the possible endangering effect of DM on the epiphyseal growth plate [58,59]. Hence, vertebral epiphyseal cross-section areas were also measured, showing the smaller dimension of the vertebral epiphyses in a portion of the diabetic rats. Despite not reaching statistical significance, the mean values of the epiphyseal cross-section areas were slightly higher in the controls than those in the diabetic animals with no clear effect of vit. K2 supplementation. Since vit. K2 supplementation started in 3-month-old animals when T2DM became manifest, the effect remained subordinate. Nevertheless, epiphysis closure is generally delayed in rats, whereby 4-month-old adult rats still show open femur epiphyses as reported previously [60], and this was also observed in our present study.

The question is whether uncontrolled T2DM affects IVD cells irreversibly so that the isolated and cultivated IVD cells might show a shift in their protein synthesis profile, which was investigated in our study. Hough et al. have undertaken an in vitro study by culturing the explanted epiphyseal cartilage slices of chronically diabetic rats in high glucose containing medium, and concluded that chronically elevated sugar levels could lead to metabolic disturbances in the epiphyseal growth plate [59]. This coincidental pathological finding seems to be in close relation to our study hypothesis, because femur length differences could result in pathological changes of the biomechanical properties of the femur. The latter is leading to gait disturbances and application of homogenous (higher) and/or heterogeneous (higher and lower) pressures in different (divergent, parallel and convergent) axes through the spine, mainly the lumbar, eventually leading to the initiation of mechanical and degenerative disc pathology. This remains a hypothesis, since we did not perform gait analysis studies prior to scarification.

Many IVDD scoring systems assessing the degenerative changes in stained histological sections were described [45,46,61].

To score the degree of IVDD, we employed the most popular scoring system of Rutges et al. because of its reliability [46]. Since this scoring system focuses on human conditions, we slightly adapted it to rat IVD histology as described in Section 2.2.5 (Materials and Methods section). The hypothesis that IVDD is associated with T2DM, discussed previously by a systematic review [15], could be confirmed by the results. AB staining performed to visualize sGAGs in situ indicated an increase of IVD in the IVDs of diabetic animals, which was in agreement with another study reporting elevated GAG content in the IVDs of diabetic mice [62].

The question was whether isolated and cultivated IVD cells from T2DM and healthy rats show an irreversible change in their typical protein expression pattern. Therefore cells were isolated and key proteins were visualized via immunolabeling of the IVD cells. The collagen family, including collagen types 1 and 2, are the most important protein components of the IVD ECM. Most IVD fibrochondrocytes were positive for collagen type 1 with a predominant intracellular and a detectable extracellular collagen expression. In addition, strong immunoreactivity for collagen type 2 was recorded for the cells negative for collagen type 1. Collagen type 1 was highly expressed in the fibrochondrocytes. There was no major difference between collagen type 1 and 2 synthesis in the cells of control and T2DM animals, but both seemed to be restored to some degree in those animals treated with vit. K2. However, collagen type 2 was induced on the gene expression level, suggesting some agreement with the study of Lintz et al. [62] which reported an increase in total collagen and GAGs in developing IVDs in diabetic mice.

SOX9 is a major chondrogenic transcription factor regulating the expression of the typical ECM markers of cartilage including collagen type 2 [63,64] and aggrecan [65]. Hence, it plays, both alone and in combination with other factors, a very important role in the process of chondrogenesis [64]. The expression of SOX9 increases during tumorigenesis of bone and cartilage and decreases in cases of cartilage disorders, such as osteoarthritis [66]. SOX9 was markedly expressed in the fibrochondrocytes derived from control and diabetic animals. It showed strong signals intracellularly (cytoplasmic) and all cells were positive for SOX9. The mean expression of SOX9 in the fibrochondrocytes derived from animals fed with additional vit. K2 was lower compared to those cells from rats without vit. K2 supplementation. This suppression was in agreement with the suppressed collagen type 1 synthesis and *aggrecan* gene expression, but contrasted with the slightly higher mean values of synthesized collagen type 2 observed in the IVD fibrochondrocytes of the rats fed with supplemental vit. K2, suggesting that vit. K2 could evoke a diverse effect on ECM composition. One has to consider as a limitation of this study that a mixed population of AF- and NP-derived cells was isolated, which would also explain the high standard deviations.

In addition to collagen, PGs are major components of the vertebral ECM. They have a considerable function in the development, tissue repair and maintenance of the vertebrae and IVDs [67]. PGs were markedly expressed in the IVD fibrochondrocytes, but not by all cells. Mostly, cytoplasmic staining showed that the PGs might remain soluble and were released into the culture supernatant. However, no major differences could be detected between groups, possibly because the antibody stains different types of PGs, which might be regulated independently.

Aggrecan is a major PG found in IVDs and the articular cartilage, playing a vital role in enhancing the mechanical, compressive and load-bearing properties of cartilage. Aggrecan loss is associated with dysfunction and degeneration of IVDs [68,69]. Aggrecan immunoreactivity was generally low in the IVD fibrochondrocytes derived from all animals. Aggrecan was more weakly expressed in cells of vit. K2-supplemented rats compared to those without vit. K2 supplementation and both, its synthesis and gene expression showed the same trend. An enhanced aggrecan degradation was mentioned by others in the IVDs of diabetic rats [13,70], but since aggrecan synthesis was intracellularly visualized in our study, we could not detect this effect.

Decorin is a small PG found in various tissues of the human body and interacts with several growth factors to control cellular vital processes, such as ECM regulation, cell-cycle and synthesis of collagen fibrils [71]. Decorin was expressed in the fibrochondrocytes derived from all animals, with no significant difference seen by comparing control and diabetic rats.

HMOX1 plays a role in tissue injury and in cartilage disorders, including IVDD [23,24,25,26,72]. It is involved in anti-oxidant, anti-apoptotic and anti-inflammatory NP cell functions, and maintenance of the regulation [73]. SOCS3 suppresses proinflammatory cytokine signaling in IVDs [27], negatively regulating the JAK/STAT signaling pathway, induced by IL-6 in lumbar discs [74]. Despite not being significant, the observed induction of *Hmox1* and *Socs3* gene expression in diabetic animals by vit. K2 might suggest the capacity to induce a response against cellular stress.

IL-10 is a member of the IL-10 cytokine family, playing an important anti-inflammatory, protective role against the hyper-/overreaction to different pathogens. In addition, it has been shown to be chondroprotective [75]. In diabetic individuals, higher plasma levels of IL-10 have been demonstrated [76]. IL-10 also has a protective influence on IVDD [77] and alleviates radicular pain [78]. Single-nucleotide or promotor-region polymorphism in the IL-10 gene was associated with IVDD in Iranian and Chinese Han populations [19,78]. In the present study, the protein expression of rat IVD cells showed no statistically significant difference, but a trend of higher IL-10 immunoreactivity in fibrochondrocytes of diabetic animals without vit. K2 supplementation.

IL–24 is a member of the same IL-10 superfamily of cytokines, expressed mainly by lymphoid and myeloid cells. It is involved in development, differentiation, inflammation, apoptosis and proliferation. IL-24 is highly expressed in rheumatoid arthritis and spondylarthritis [79], but its presence and role in IVDs and IVD-derived fibrochondrocytes is completely unclear. Interestingly, we could detect it in cultured IVD cells. The cells showed mostly weak and cytoplasmic signals for IL-24. Obviously, it is released in vesicles, likely representing exosomes. However, its role remains unclear and there were no significant differences in the cells of non-diabetic and diabetic animals.

αSMA is the actin isoform widely distributed in vascular smooth muscle cells and effectively participates in fibrogenesis. Activated fibroblasts can produce αSMA, which is strongly related to the fibrotic reaction [80] and fibrotic alterations in the AF [81]. Regardless of origin all cells showed intensive intracellular signals for αSMA. Previously, it was shown that fibrochondrocytes, in this case derived from the meniscus, could indeed undergo myofibroblast transition [82] as seen for fibroblasts. αSMA was highly expressed in cultured fibrochondrocytes derived from non-diabetic and diabetic animals. Without vit. K2 supplementation, the expression level was almost equal in the IVD fibrochondrocytes, while its expression was obviously increased by vit. K2 in both groups (controls and T2DM). Gene expression of cytoprotective complement regulatory proteins *Cd55* and *Cd59* revealed an inductive trend in diabetic animals receiving vit. K2 supplementation.

## 5. Conclusions

The lumbar motion segments, particularly the IVDs, can mirror the effect of T2DM on musculoskeletal tissues. The femoral lengths of diabetic rats were significantly shorter than those in the control group, which could be related to a devastating or inhibitory effect of T2DM on the epiphyseal growth plates (EGPs). Epiphysis sizes followed a similar trend. Vit. K2 did not show any protective effect in epiphyseal growth plates, possibly due to the late period of vit. K2 supplementation. T2DM could influence the regulation of ECM components (*aggrecan* and *collagen type 2*). Vit. K2 might counteract the inflammatory process of IVDD by regulating *Socs3* and *Hmox1*. To elucidate the concrete involvement of the investigated cytokines (IL-10 and IL-24) in IVDD of diabetic individuals, further investigation is required.

Finally, T2DM could be considered as a primary cause of initiation or a secondary cause of exaggerating the occurrence of IVDD, since we observed significantly higher IVD degeneration in the diabetic compared to the non-diabetic rats. Vit. K2 may exert some protective effects which require further investigation.

## Figures and Tables

**Figure 1 nutrients-15-02872-f001:**
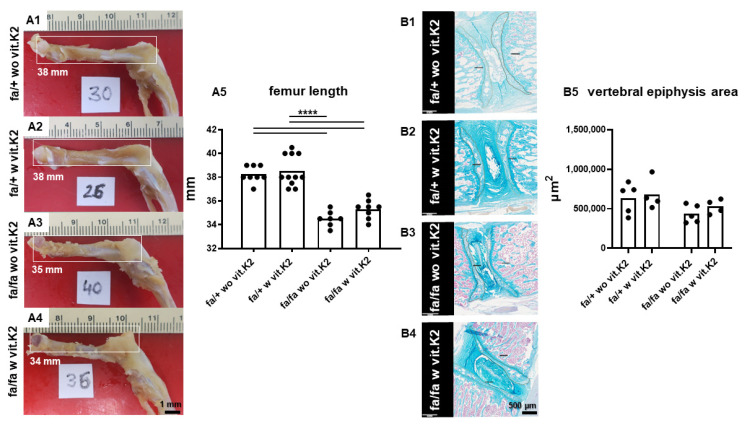
Femoral lengths and vertebral epiphyseal cross-section areas of the non-diabetic and diabetic ZDF rats without and with additional vit. K2 feeding. (**A1**–**A3**): Representative view on the left femurs of the four subgroups (the length in mm is inserted, the number in each photo means the animal number): non-diabetic ZDF rats without vit. K2 (fa/+ wo vit.K2, **A1**) and with vit. K2 (fa/+ w vit.K2, **A2**), diabetic ZDF rats without vit. K2 (fa/fa wo vit.K2, **A3**) and with vit. K2 (fa/fa w vit.K2, **A4**). (**A5**): Summary of measurements of femur lengths of the four groups. (**B1**–**B5**): Measurement of the epiphyseal cross-sectional area of IVDs (encircled) in the different groups’ summary of measurements (**B5**). Outlier test: ROUT (Q = 1%). One-way ANOVA (multiple comparison test). *p* values **** < 0.001. Tukey post hoc test. fa/+: heterozygous and healthy rats; fa/fa: homozygous and diabetic rats. Scale bar: 1 mm (**A1**–**A4**), 0.5 mm (**B1**–**B4**). (**A5**): n = 7–11, (**B5**): n = 4–5.

**Figure 2 nutrients-15-02872-f002:**
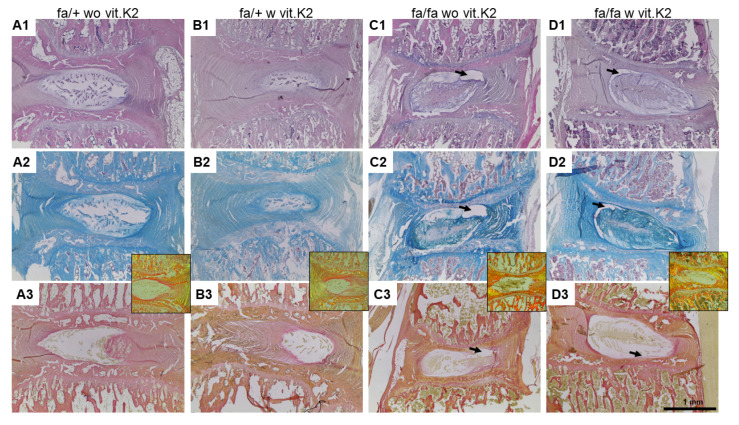
Histological staining of explanted rat lumbar intervertebral discs (IVD). Representative histological IVD sections of non-diabetic ZDF rats (fa/+, **A**,**B**) or diabetic ZDF rats (fa/fa, **C**,**D**), fed without vitamin K2 (wo vit.K2, **A**,**C**) or with vitamin K2 (w vit.K2, **B**,**D**) supplementation. The sections were stained with Hematoxylin–Eosin stain (**A1**–**D1**) to acquire information on the overall histological structure, Alcian blue (**A2**–**D2**) for sulfated glycosaminoglycan detection and Sirius red stain (**A3**–**D3**) for collagen structure depiction. Insets show Sirius red stain visualized with polarized light. Scale bar: 1 mm. Arrows: showing cleft formation around the NP.

**Figure 3 nutrients-15-02872-f003:**
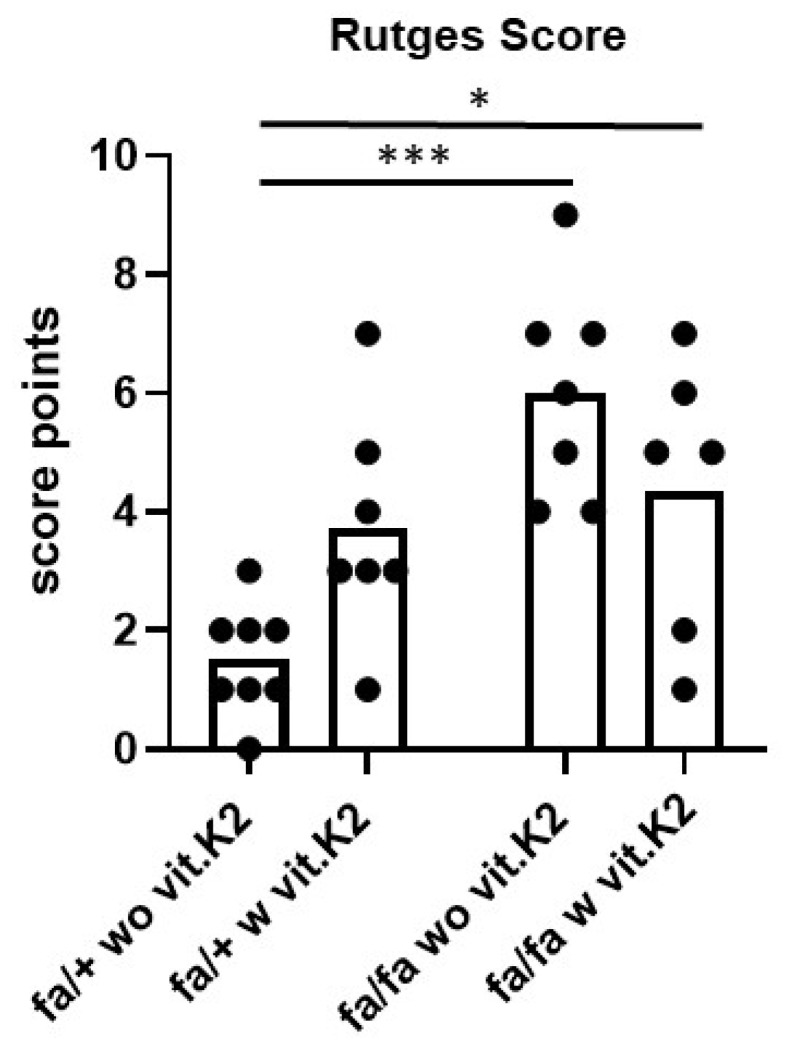
Intervertebral disc (IVD) degeneration of rats. For IVD scoring, the adapted Rutges score was applied [46]. Non-diabetic ZDF rats fed without vit. K2 (fa/+ wo vit.K2) and with vit. K2 (fa/+ w vit.K2), diabetic ZDF rats fed without vit. K2 (fa/fa wo vit.K2) and with vit. K2 (fa/fa w vit.K2) supplementation. Outlier test: ROUT (Q = 1%). One-way ANOVA. Tukey post hoc test. *p* values * < 0.05, *** < 0.005. fa/+: heterozygous and healthy rats; fa/fa: homozygous and diabetic rats. n = 6–8.

**Figure 4 nutrients-15-02872-f004:**
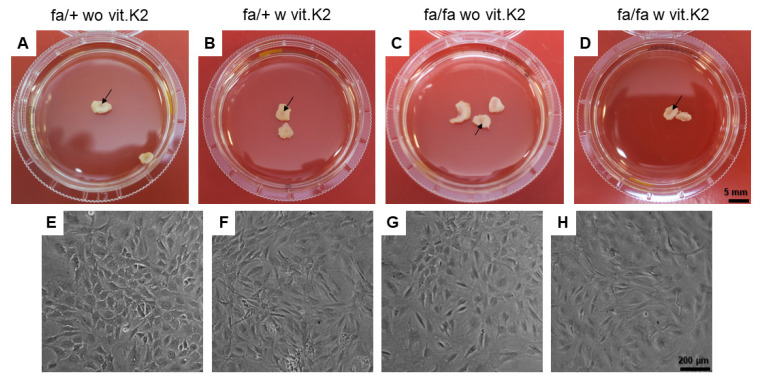
Explanted intervertebral discs (IVDs) for IVD cell isolation. Lumbar IVDs explanted for cell isolation (**A**–**D**, arrows: nucleus pulposus) are shown and cultivated IVD fibrochondrocytes (**E**–**H**) derived from the four rat groups of the study. Non-diabetic ZDF rats fed without additional vit. K2 (fa/+ wo vit.K2) and with vit. K2 (fa/+ w vit.K2), diabetic ZDF rats fed without vit. K2 (fa/fa wo vit.K2) and with vit. K2 (fa/fa w vit.K2) supplementation. Cells depicted (**E**–**H**) derived from passage 0–1. Scale bars: 5 mm (**A**–**D**), 200 µm (**E**–**H**).

**Figure 5 nutrients-15-02872-f005:**
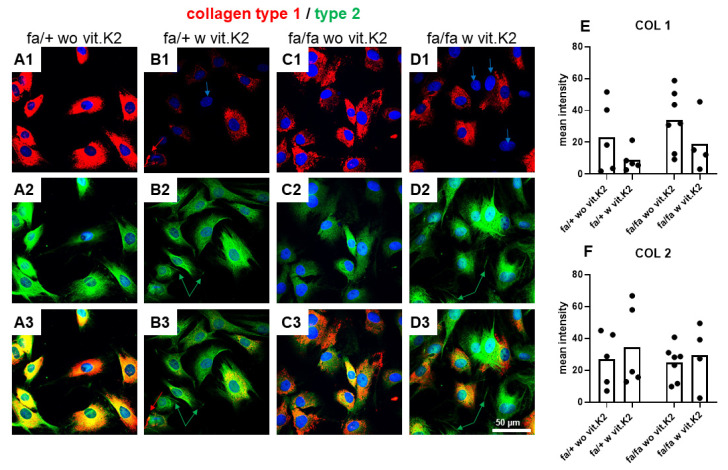
Collagen type 1 and 2 expression in intervertebral disc (IVD) cells of non-diabetic (fa/+) and diabetic (fa/fa) ZDF rats fed without (wo vit.K2) or with vitamin K2 (w vit.K2) supplementation. (**A1**–**D1**): collagen type 1 (red, blue arrows indicate negative cells), (**A2**–**D2**): collagen type 2 (green). Cell nuclei are counterstained with 4′,6-diamidino-2-phenylindole (DAPI, blue). (**A3**–**D3**): overlay. (**B1**,**B3**): red arrows show extracellular collagen type 1; the green arrows show extracellular collagen type 2 (**B2**,**B3**,**D2**,**D3**). Densitometric evaluation of collagen type 1 (COL 1, **E**) and collagen type 2 (COL 2, **F**) immunoreactivity. Outlier test: ROUT (Q = 1%). One-way ANOVA. Tukey post hoc test. fa/+: heterozygous and healthy rats; fa/fa: homozygous and diabetic rats. Scale bars: 50 µm. n = 4–7.

**Figure 6 nutrients-15-02872-f006:**
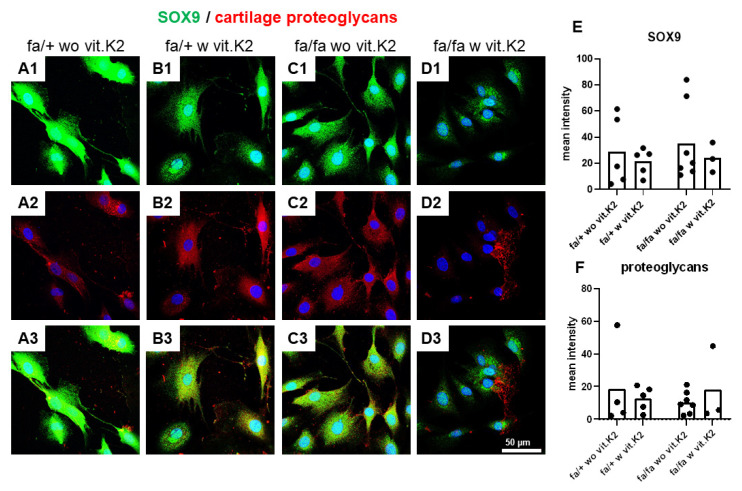
SOX9 and cartilage proteoglycans expression in intervertebral disc (IVD) cells of non-diabetic (fa/+) and diabetic (fa/fa) rats fed without (wo vit.K2) or with vitamin K2 (w vit.K2) supplementation. (**A1**–**D1**): SOX9 (green), (**A2**–**D2**): cartilage proteoglycans (red). Cell nuclei are counterstained with 4′,6-diamidino-2-phenylindole (DAPI, blue). (**A3**–**D3**): overlay. Densitometric evaluation of SOX9 (**E**) and proteoglycan immunoreactivity (**F**). Non-diabetic ZDF rats without vit. K2 (fa/+ wo vit.K2) and with vit. K2 (fa/+ w vit.K2), diabetic ZDF rats without vit. K2 (fa/fa wo vit.K2) and with vit. K2 (fa/fa w vit.K2). Outlier test: ROUT (Q = 1%). One-way ANOVA. Tukey post hoc test. Scale bars: 50 µm. n = 3–7.

**Figure 7 nutrients-15-02872-f007:**
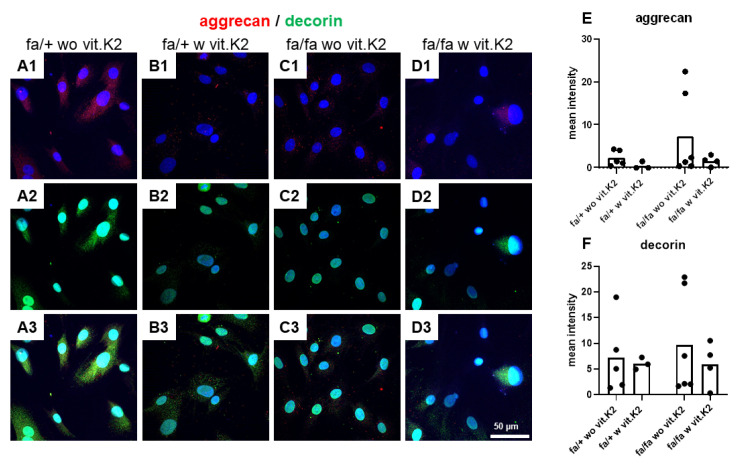
Aggrecan and decorin expression in intervertebral disc (IVD) cells of non-diabetic (fa/+) and diabetic (fa/fa) ZDF rats without (wo vit.K2) or with vitamin K2 (w vit.K2) supplementation**.** (**A1**–**D1**): aggrecan (red), (**A2**–**D2**): decorin (green). Cell nuclei are counterstained with 4′,6-diamidino-2-phenylindole (DAPI, blue). (**A3**–**D3**): overlay. Densitometric evaluation of aggrecan (**E**) and decorin (**F**) immunoreactivity. Outlier test: ROUT (Q = 1%). One-way ANOVA. Tukey post hoc test. Scale bars: 50 µm. fa/+: heterozygous and healthy rats; fa/fa: homozygous and diabetic rats. n = 3–6.

**Figure 8 nutrients-15-02872-f008:**
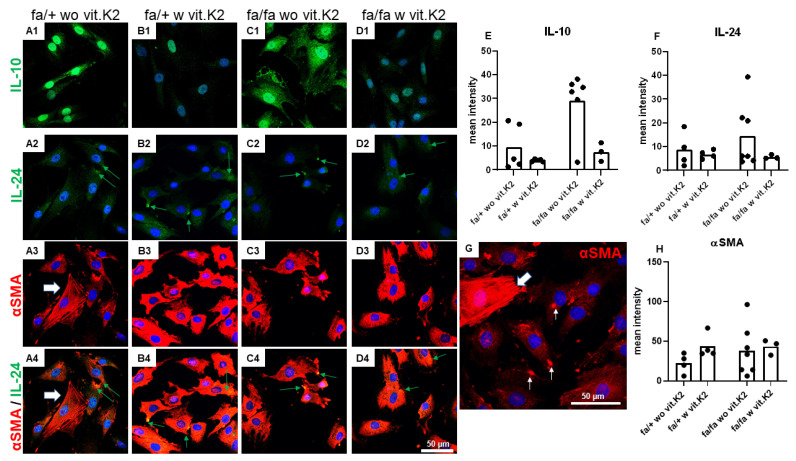
IL-10, IL-24 and αSMA expression in intervertebral disc (IVD) cells of non-diabetic (fa/+) and diabetic (fa/fa) ZDF rats fed without (wo vit.K2) and with (w vit.K2) additional vitamin K2. (**A1**–**D1**): IL-10 (green), (**A2**–**D2**): IL-24 (green), (**A3**–**D3**): alpha smooth muscle actin (αSMA, red). Cell nuclei are counterstained with 4′,6-diamidino-2-phenylindole (DAPI, blue). (**A4**–**D4**): overlay of IL-24 and αSMA stain. **G**: higher magnification of an αSMA stain (fa/fa wo vit.K2). White thick arrow (**A3**,**G**): cells with αSMA-positive stress fibers. Thin white arrows: highly SMA-positive intracellular plaques (**G**). Green arrows (**A2**–**D2**,**A4**–**D4**): vesicles more strongly immunoreactive for IL-24. Densitometric evaluation of immunoreactivity of IL-10 (**E**), IL-24 (**F**) and αSMA (**H**). Outlier test: ROUT (Q = 1%). One-way ANOVA. Tukey post hoc test. Scale bars: 50 µm. fa/+: heterozygous and healthy rats; fa/fa: homozygous and diabetic rats, IL: interleukin. n = 3–7.

**Figure 9 nutrients-15-02872-f009:**
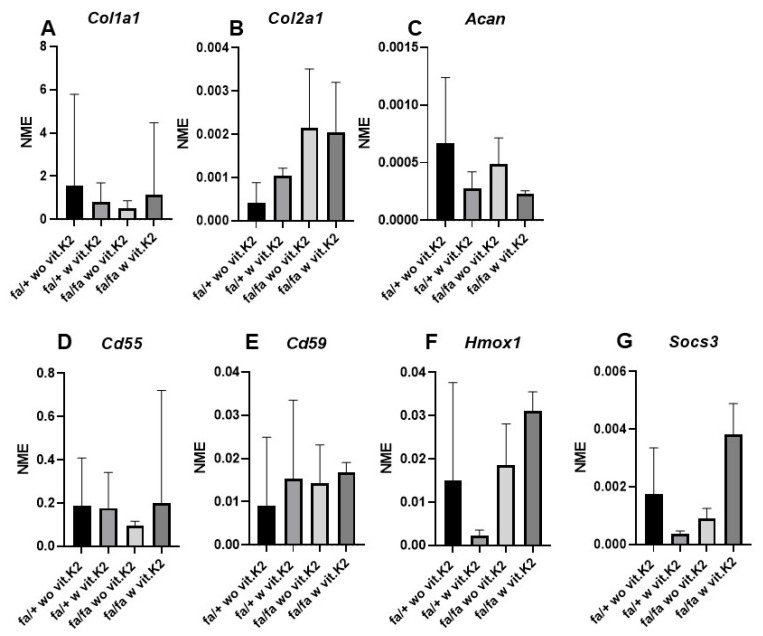
Mean normalized gene expression of *collagen type 1* (*Col1a1*, alpha chain, (**A**)), *type 2* (*Col2a1*, alpha chain (**B**)), *aggrecan* (*Acan,* (**C**)), *Cd55* (**D**), *Cd59* (**E**), *heme oxygenase (Hmox*)*-1* (**F**) and *suppressor of cytokine signaling* (*Socs*)*3* (**G**) are shown in intervertebral disc (IVD) cells of non-diabetic (fa/+) and diabetic (fa/fa) ZDF rats fed without (wo vit.K2) and with (w vit.K2) additional vitamin K2. Outlier test: ROUT (Q = 1%). One-way ANOVA. Tukey post hoc test.

**Table 1 nutrients-15-02872-t001:** Experimental animal groups.

Non-Diabetic Rats		Diabetic Rats	
fa/+ without vit. K2 (fa/+ wo vit.K2)	fa/+ with vit. K2 (fa/+ w vit.K2)	fa/fa without vit. K2 (fa/fa wo vit.K2)	fa/fa with vit. K2 (fa/fa w vit.K2)
9 rats	11 rats	7 rats	9 rats

Fa/+: heterozygous ZDF rat, fa/fa: homozygous ZDF rat, vit.: vitamin, w: with, wo: without, T2DM: diabetes mellitus type 2.

**Table 2 nutrients-15-02872-t002:** Modified histological IVD degeneration score based on [46] Rutges et al. (2013).

Item and Grade		0	1	2
AF		Normal lamellar pattern	Slight lamellar disorganization	Loss of lamellar organization
AF/NP-border		Normal	Minimal interruption, distinct space/bleb formation between both	Loss of distinction, severe space/bleb formation between both
NP cellularity		Normal cellularity and arrangement	Mild cell density or distribution changes, few cell clusters	Profound cell density/distribution changes, prominent cell clusters
NP ECM distribution		Normal gelatinous appearance	Mild condensation, disorganization	Severe condensation, disorganization
NP morphology	NP shape	Oval	Oval/round, mild distortion	Irregular shape
	NP area	NP constitutes more than 50% of disc area	NP constitutes between 25–50% of disc area	NP constitutes less than 25% of disc area
CEP		No bony sclerosis and regular CEP	Mild bony sclerosis with mildly thickened CEP	Severely thickened CEP with severe sclerosis
Maximal number of points (severe degeneration): 14				

AF: annulus fibrosus; CEP: cartilaginous endplate; ECM: extracellular matrix, NP: nucleus pulposus.

**Table 3 nutrients-15-02872-t003:** Antibodies and staining used to assess protein expression in cells.

Target	Primary Antibody	Dilution	Secondary Antibody	Dilution
Aggrecan	mouse-anti-human, R&D systems, Minneapolis, MN, USA	1:30	donkey-anti-mouse; Cy3, Invitrogen, Carlsbad, CA, USA	1:200
Collagen type 1	goat-anti-human, (COL1A1 chain), Abcam, Cambridge, UK	1:50	donkey-anti-goat; Cy3, Invitrogen, Carlsbad, CA, USA	1:200
Collagen type 2	rabbit-anti-human, Acris Laboratories, Hiddenhausen, Germany	1:50	donkey-anti-rabbit, Alexa Fluor 488, Invitrogen, Carlsbad, CA, USA	1:200
Decorin	rabbit-anti-human, OriGene Rockville, MD, USA	1:50	donkey-anti-rabbit; Alexa-Fluor488, Invitrogen, Carlsbad, CA, USA	1:200
IL-10	rabbit-anti-human, PeproTech, Cranbury, NJ, USA	1:30	donkey-anti-rabbit, Alexa Fluor 488, Invitrogen, Carlsbad, CA, USA	1:200
IL-24	rabbit-anti-human, Invitrogen, Carlsbad, CA, USA	1:30	donkey-anti-rabbit; Alexa-Fluor488, Invitrogen, Carlsbad, CA, USA	1:200
IL-24	mouse-anti-human, R&D systems, Minneapolis, MN, USA	1:30	donkey-anti-mouse; Cy3, Invitrogen, Carlsbad, CA, USA	1:200
Phalloidin Alexa488	Santa Cruz Biotechnologies, Santa Cruz, CA, USA	1:100	none	
Proteoglycans	mouse-anti-human, Chemicon International, Temecula, CA, USA	1:70	donkey-anti-mouse; Cy3, Invitrogen, Carlsbad, CA, USA	1:200
SOX9	rabbit-anti-human, Merck, Darmstadt, Germany	1:100	donkey-anti-rabbit, Alexa Fluor 488, Invitrogen, Carlsbad, CA, USA	1:200
αsmooth muscle actin	mouse-anti-human, Sigma-Aldrich (A5228), Munich, Germany	1:50	donkey-anti-mouse; Cy3, Invitrogen, Carlsbad, CA, USA	1:200

Cy3: cyanine 3, COL1A1: collagen type 1 alpha1 chain, IL: interleukin, SOX9: SRY-box transcription factor 9.

**Table 4 nutrients-15-02872-t004:** Rat-specific primers used for RTD PCR in this study.

Gene Symbol	Gene Name	Assay ID	Amplicon Length (bp)	Efficiency	NCBI Gene Reference
*Acan* *	*Aggrecan*	Rn00573424_m1	74	2.04	NM_022190.1
*Cd55*	*Decay accelerating factor*	Rn00709472_m1	92	1.94	NM_022269.2
*Cd59*	*Protectin/MAC inhibitory protein*	Rn00563929_m1	80	2.05	NM_012925.1
*Col1a1*	*Collagen type 1, alpha 1*	Rn01463848_m1	115	2.01	NM_053304.1
*COL2A1* **	*Collagen type 2, alpha 1*	Ec03467411_m1	81	2.17	NM_001081764.1
*Hmox1*	*Heme oxygenase-1*	Rn00561387_m1	132	2.25	NM_012580.2
*Hprt1*	*Hypoxanthine-Guanine-phosphoribosyltransferase*	Rn01527840_m1	64	1.98	NM_012583.2
*Socs3*	*Suppressor of Cytokine Signaling 3*	Rn00585674_s1	73	1.90	NM_053565.1

* All primers were obtained from ThermoFisher Scientific (Germany), bp: base pairs. ** this equine primer displayed high efficacy for rat samples and hence, could be used.

## Data Availability

Data can be shared on request.

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
