# Peer review of "Does Vitamin K2 Influence the Interplay between Diabetes Mellitus and Intervertebral Disc Degeneration in a Rat Model?"

_nutrients, 2023, doi:10.3390/nu15132872_

Round 1
Reviewer 1 Report
The manuscript aimed to investigate the beneficial effects of K2 on intervertebral disc (IVD) degeneration in Zucker Diabetes Fatty rats and investigation of some signaling pathways. In my opinion, the manuscript is prepared well.
Major comments
1. The Introduction is too lengthy.
2. The results should be presented in scatter plots, not floating bars.
3. Authors should try to validate their data using Western blot or flow cytometry in the experiments presented in Figure 5, 6, 7, and 8.
Additional minor concerns
1. What was the formula used to normalize the PCR data? Was it the 2-deltadeltaCt method?
2. Authors should provide references on how the dose of vitamin K2 used in the study was decided upon
3. As they are assessing gene expression, gene name and not protein name should be used in Fig 9.
The author has good English writing ability.
However, some suggestions for the structure of the article are as follows
The introduction and discussion sections are excessively lengthy and would benefit from a more concise approach.
It is recommended to focus on the main content of the study, eliminating unnecessary details that may divert the reader's attention.
Author Response
Dear Ladies and Gentlemen, Nuremberg, 12th June 2023
Dear Editor,
The authors would like to thank the reviewers for carefully reading the manuscript and their very valuable comments. We modified the manuscript according to the reviewers suggestions with a list of changes shown below. All corrections and addenda performed are indicated in red in the revised version of the manuscript. We hope you will find this manuscript suitable for publication in “Nutrients”. Please do not hesitate to contact me anytime for questions regarding this manuscript. Several orthographic mistakes were corrected and figures were optimized.
Sincerely,
Univ.-Prof. Dr. Gundula Schulze-Tanzil
As requested by Crey Zhao we added a graphical abstract now as a line drawing without photos to make it more attention grabbing.
Reviewer 1
The manuscript aimed to investigate the beneficial effects of K2 on intervertebral disc (IVD) degeneration in Zucker Diabetes Fatty rats and investigation of some signaling pathways. In my opinion, the manuscript is prepared well.
Response: We thank reviewer 1 for his/her encouraging comment.
Is the research design appropriate? “can be improved.”
Response: We improved the research design by increasing the number of independent experiments with IVD cells of 2-4 additional donors in the diabetic group (immunofluorescence staining). We added two independent experiments with IVD cells of 2 additional diabetic donors in the PCR setting. Due to cell and time limitations (14 days revision time) further exeriments were not possible.
Are the methods adequately described? “can be improved.”
Response: We improved description of methods e.g. section 2.3.4
Are the results clearly presented? “can be improved.”
Response: We improved presentation and discription of results using scatter plots.
Are the conclusions supported by the results? “can be improved.”
Response: The conclusion section was revised.
Major comments
- The Introduction is too lengthy.
Response: We shortened it substantially now.
- The results should be presented in scatter plots, not floating bars.
Response: We transformed the diagrams in scatter plots.
- Authors should try to validate their data using Western blot or flow cytometry in the experiments presented in Figure 5, 6, 7, and 8.
Response: It is not possible to perform two novel methods within 10 days of revision time. We tried to add experimental data from IVD cells derived from additional donors for immunofluorescence and RTD-PCR, but were limited by time and cells, since we could not isolated from every animal sufficient cells. The data are included now in Figures 5-9.
Additional minor concerns
- What was the formula used to normalize the PCR data? Was it the 2-deltadeltaCt method?
Response: It is the Delta CT method described by Schefe et al., was used here. Accordingly, we cite this reference now: Schefe, J.H.; Lehmann, K.E.; Buschmann, I.R.; Unger, T.; Funke-Kaiser, H. Quantitative real-time RT-PCR data analysis: current concepts and the novel "gene expression's CT difference" formula. J Mol Med (Berl) 2006, 84, 901-910, 1004. doi:10.1007/s00109-006-0097-6.
- Authors should provide references on how the dose of vitamin K2 used in the study was decided upon
Response: In this work we used vitamin K2 (MK-7) in the concentration of 100 mg/kg of feed likewise, as it was performed in the following manuscripts:
- Zaragatski E, Grommes J, Schurgers LJ, Langer S, Kennes L, Tamm M, Koeppel TA, Kranz J, Hackhofer H, Arakelyan K, Jacobs MJ, Kokozidou Vitamin K antagonism aggravates chronic kidney disease-induced neointimal hyperplasia and calcification in arterialized veins: role of vitamin K treatment? Kidney Int 2016; 89(3):601-11. doi: 10.1038/ki.2015.298. PMID: 26466318
- Scheiber D, Veulemans V, Horn P, Chatrou ML, Potthoff SA, Kelm M, Schurgers LJ, Westenfeld High-Dose Menaquinone-7 Supplementation Reduces Cardiovascular Calcification in a Murine Model of Extraosseous Calcification. Nutrients 2015; 7(8):6991-7011. doi: 10.3390/nu7085318. PMID: 26295257
- Schurgers LJ, Spronk HMH, SouteBAM, Schiffers PM, DeMey JGR, VermeerRegression of warfarin-induced medial elastocalcinosis by high intake of vitamin K in rats. Blood 2007; 109(7):2823-31. doi: 10.1182/blood-2006-07-035345. PMID: 17138823
The above publications used 100 mg/kgr vit.K2-MK4 in the rat feed. It is known from the literature that vit.K2-MK7 which is the one we used, has even more enhanced properties in vasculature, bone strength and collagen production from osteoblasts (Sato, T., N. Inaba, and T. Yamashita, MK-7 and Its Effects on Bone Quality and Strength. Nutrients, 2020. 12:4) compared to vit.K2-MK4. Therefore, we expect that by using the above concentration we should have stable results.
Additionally, Dr. Mrosewski measured Vitamins K1, K2-MK4 and K2-MK7 in the final serum of the sacrificed animals after the 80 days of vit. K2-MK7 feeding. The average values are summarized in the figure. The animal groups, both control and T2DM that were supplemented with vit.K2-MK-7 present high vit.K2-MK7 values in their final serum. The vitamin K2 data together with other serum analyses are part of another publication (not yet published) and hence, cannot be added to this manuscript.
|
In the serum of the ZDF rats vitamine K2-MK4 and viamine K2-MK7 was measured. Significantly higher levels of vitamine K2-MK7 could be found in the rats feed with vit. K2 compared to the rats without vit. K2 feeding and also higher levels in diabetic compared to non diabetic animals. |
|
- As they are assessing gene expression, gene name and not protein name should be used in Fig 9.
Response: We added the gene names in cursive throughout the manuscript.
However, some suggestions for the structure of the article are as follows
The introduction and discussion sections are excessively lengthy and would benefit from a more concise approach.
Response: We shortened it substantially now.
It is recommended to focus on the main content of the study, eliminating unnecessary details that may divert the reader's attention.
Response: We shortened it substantially now.

Reviewer 2 Report
Dear Authors,
Your publication concerns a very important problem of intervertebral disc degeneration (IVD) in patients with type 2 diabetes (T2DM) and the impact of vitamin (vitamin) K2 on the symptoms and pharmacological and physiological effects observed during this disease. This paper is written very interestingly and correctly. The methodology and results are presented concisely. The discussion was based on the latest literature data, presents current research problems in the field of understanding the patho-mechanism of the development of complications in type 2 diabetes. I noticed a few minor editorial remarks, the improvement of which, in my opinion, will contribute to a better quality of your publication:
Minor notes:
1. Table 1 and Table 3 - explain abbreviations
2. Table 2 - larger spacing between table columns, the text in the table is illegible because it is packed too tightly
3. Figure 1 - very small inscriptions next to the graphs, I suggest to increase the font size
4. #533 and #534 - this information should not be in the "Results" section, I suggest moving this sentence with the citation to the "Introduction" or "Discussion" section
5. #550-559 - this description should be in the "Results" section, and in the "Discussion" section only refer to this information
6. #566 - please cite where this information comes from?
7. #577 - too long space between sentences
8. #642 - the subsection number is given, please also write the name of this subsection: "Materials and methods"
9. #766 - too much space between sentences and there are 2 dots, please remove one
Author Response
Dear Ladies and Gentlemen, Nuremberg, 12th June 2023
Dear Editor,
The authors would like to thank the reviewers for carefully reading the manuscript and their very valuable comments. We modified the manuscript according to the reviewers suggestions with a list of changes shown below. All corrections and addenda performed are indicated in red in the revised version of the manuscript. We hope you will find this manuscript suitable for publication in “Nutrients”. Please do not hesitate to contact me anytime for questions regarding this manuscript. Several orthographic mistakes were corrected and figures were optimized.
Sincerely,
Univ.-Prof. Dr. Gundula Schulze-Tanzil
As requested by Crey Zhao we added a graphical abstract now as a line drawing without photos to make it more attention grabbing.
Reviewer 2
Are the results clearly presented: “can be improved.”
Response: We improved presentation and description of results.
Dear Authors, Your publication concerns a very important problem of intervertebral disc degeneration (IVD) in patients with type 2 diabetes (T2DM) and the impact of vitamin (vitamin) K2 on the symptoms and pharmacological and physiological effects observed during this disease. This paper is written very interestingly and correctly. The methodology and results are presented concisely. The discussion was based on the latest literature data, presents current research problems in the field of understanding the patho-mechanism of the development of complications in type 2 diabetes. I noticed a few minor editorial remarks, the improvement of which, in my opinion, will contribute to a better quality of your publication:
Response: We thank the reviewer for his/her encouraging comments.
Minor notes:
- Table 1 and Table 3 - explain abbreviations
Response: We added explanations of the abbreviations to both tables.
- Table 2 - larger spacing between table columns, the text in the table is illegible because it is packed too tightly.
Response: We adapted table 2, deleted the empty column for “points”, shortened text and generated more spaces between rows and columns.
- Figure 1 - very small inscriptions next to the graphs, I suggest to increase the font size
Response: We adapted it.
- #533 and #534 - this information should not be in the "Results" section, I suggest moving this sentence with the citation to the "Introduction" or "Discussion" section
Response: We deleted this explanation in the result section.
- #550-559 - this description should be in the "Results" section, and in the "Discussion" section only refer to this information
Response: We transferred this clinical observation into the result section.
- #566 - please cite where this information comes from?
Response: In line with the demand to shorten the discussion we removed this hypothetical statement..
- #577 - too long space between sentences
Response: Now, line 542, we shortened the space.
- #642 - the subsection number is given, please also write the name of this subsection: "Materials and methods"
Response: We added it.
- #766 - too much space between sentences and there are 2 dots, please remove one
Response: We adapted it.

Round 2
Reviewer 1 Report
Authors have successfully addressed most of the reviewers comments, and I feel that the MS is now acceptable for publication.
The conclusion of the article is accurately described and can be published with appropriate language modifications.